# HPV16 Load Is a Potential Biomarker to Predict Risk of High-Grade Cervical Lesions in High-Risk HPV-Infected Women: A Large Longitudinal French Hospital-Based Cohort Study

**DOI:** 10.3390/cancers13164149

**Published:** 2021-08-18

**Authors:** Antoine Baumann, Julie Henriques, Zohair Selmani, Aurélia Meurisse, Quentin Lepiller, Dewi Vernerey, Séverine Valmary-Degano, Sophie Paget-Bailly, Didier Riethmuller, Rajeev Ramanah, Christiane Mougin, Jean-Luc Prétet

**Affiliations:** 1Papillomavirus National Reference Centre, CHU de Besançon, 25000 Besançon, France; antoine.baumann@ch-colmar.fr (A.B.); zselmani@chu-besancon.fr (Z.S.); quentin.lepiller@univ-fcomte.fr (Q.L.); DRiethmuller@chu-grenoble.fr (D.R.); christiane.mougin@univ-fcomte.fr (C.M.); 2Pathology Department, CHU de Besançon, 25000 Besançon, France; svalmarydegano@chu-grenoble.fr; 3Methodology and Quality of Life Unit in Oncology, CHU de Besançon, 25000 Besançon, France; jhenriques@chu-besancon.fr (J.H.); ahusse@chu-besancon.fr (A.M.); dvernerey@chu-besancon.fr (D.V.); spaget@chu-besancon.fr (S.P.-B.); 4UMR1098, Host-Graft-Tumor Interactions and Cell and Tissue Engineering, University Bourgogne Franche-Comté, INSERM, 25000 Besançon, France; 5La Fédération Hospitalo-Universitaire Integrated Center for Research in Inflammatory Diseases (FHU Increase), ANR-11-LABX-0021, LabEx LipSTIC, University Bourgogne Franche-Comté, 25000 Besançon, France; 6EA3181, University Bourgogne Franche-Comté, 25000 Besançon, France; 7Gynecology Department, CHU de Besançon, 25000 Besançon, France; rajeev.ramanah@univ-fcomte.fr

**Keywords:** HPV16 viral load, prognosis, cervix, high-grade lesion

## Abstract

**Simple Summary:**

This study aimed at assessing HPV16 and HPV18 viral loads to predict the development of cervical high-grade lesion. Among 885 women positive for hrHPV and presenting no or mild cytological abnormalities, HPV16 and HPV18 prevalence was 25.9% and 8.4%, respectively. Of those women, 135 developed a high-grade lesion during the follow-up. Considering an HPV16 viral load cut-off set at 3.2 log_10_ GE/10^3^ cells a subgroup of women at high risk of developing high-grade cervical lesion (HR = 2.67; 95% CI 1.80–3.97 *p* ≤ 0.0001) has been identified. Moreover, a composite score based on HPV16 load, cytology and hrHPV detection allowed for CIN2+ risk stratification. To conclude, HPV16 load is a relevant biomarker to identify women at high risk for developing precancerous lesions of the cervix.

**Abstract:**

High-risk HPV (hrHPV) testing has been implemented as a primary screening tool for cervical cancer in numerous countries. However, there is still a need for relevant triage strategies to manage hrHPV positive women to avoid excessive referral to colposcopy. The objective of this study was to assess, in women infected by hrHPV and presenting no or mild cytological abnormalities, HPV16 and HPV18 viral loads to predict the development of cervical high-grade lesion. Among 2102 women positive for hrHPV, 885 had no lesion or mild cytological abnormalities at baseline and had at least one follow-up (FU) visit. HPV16 and HPV18 prevalence was 25.9% and 8.4%, respectively. Of those women, 15% developed a high-grade lesion during the FU. An HPV16 viral load cut-off set at 3.2 log_10_GE/10^3^ cells permitted to identify a subgroup of women at high risk of developing high-grade cervical lesion (HR = 2.67; 95% CI 1.80–3.97; *p* ≤ 0.0001). No specific HPV18 viral load threshold could have been defined in regard to the present study. In multivariate analysis, HPV16 load (absence/log_10_GE/10^3^ cells < 3.2 vs. ≥3.2), RLU/PC 239 (1–100 pg/mL vs. >100 pg/mL) and cytology (normal vs abnormal) were independently associated with a significant increased risk of high-grade lesion development and were used to construct the prognostic score. In conclusion, HPV16 load is a relevant biomarker to identify women at high risk for developing cervical precancerous lesions.

## 1. Introduction

High-risk HPVs (hrHPV) have been recognized as the etiologic agents of cervical cancer [1,2]. However, only persistent infections confer an increased risk of developing precancerous and cancerous lesions [3,4]. Among the twelve HPV genotypes having oncogenic properties, HPV16 is the most at risk, causing cancer of the cervix and at several other sites [5]. In addition, HPV16, HPV18 is the second most prevalent hrHPV in cervical cancer [6,7,8] and both HPV types have been associated with a high ten-year risk of cervical intraepithelial neoplasia grade 2 or worse (CIN2+) [9].

Cervical cancer screening has been based on cytological analysis of cervical smears since the 1950s. Two decades ago, hrHPV testing has been proposed as a potential alternative to repeated cytology or immediate colposcopy for the triage of women with Atypical Squamous Cells of Undetermined Significance (ASC-US) cytology [10]. In the few last years, hrHPV testing has also been introduced in the management protocols of women treated for CIN and it is now considered the standard test of cure after excisional treatment of cervical high-grade lesion [11]. Recently, a Cochrane review demonstrated the superiority of hrHPV testing compared to cytology to detect high-grade lesions [12]. Thus, several countries have changed their screening policies by switching from cytology to primary hrHPV testing [13].

However, the authors of the Cochrane’s review pointed out that relevant triage strategies are needed to manage hrHPV positive women [12]. For instance, reflex cytology is an approach adopted by different countries including France. HPV16 and HPV18 genotyping is another option for stratification of hrHPV positive women. In this line, the ATHENA HPV study showed that the relative risks for CIN3+ in women infected by HPV16 or HPV18 were, respectively, 42.0% and 20.5% compared with hrHPV negative women. Moreover, the relative risk in women infected by non-16 and non-18 hrHPV was only 8%. This is why it has been proposed to refer HPV16/18 positive women with normal cytology to immediate colposcopy [14,15].

Other biomarkers have been assessed to manage hrHPV positive women that include HPV genotyping, p16/Ki67 dual-staining, or methylation status of HPV and some human genes [16,17,18,19,20]. Assessing HPV16/18 viral load (VL) could also represent an attractive option and few longitudinal studies showed that HPV load levels might reflect the natural history of cervical lesions [21,22,23].

In the present study, HPV16 and HPV18 VL were assessed as potential risk factors for the development of high-grade lesions from a large French hospital-based cohort of women infected by hrHPV and presenting without or with equivocal or mildly abnormal cytology.

## 2. Materials and Methods

### 2.1. Patients

From January 2009 to September 2017, women attending the Gynecology Clinic of the Besançon’s University Hospital consented to cervical cancer screening with conventional Pap cytology and hrHPV test using the Hybrid Capture 2 (hc2) assay (Qiagen, Courtaboeuf, France). They were followed-up as previously described [24]. Cervical smears positive for hrHPV were further tested for HPV16 and HPV18 VL, the two most prevalent HPV in cervical cancers, using homemade real-time PCRs [25]. Women presenting cervical abnormalities (cytology ≥ LSIL) at the first visit or during the FU were referred for colposcopy and managed according to the French clinical practice guidelines.

The main study endpoint was the histological diagnosis of a high-grade lesion during the FU. In this way, women presenting a high-grade lesion of the cervix (HSIL cytology or CIN2+ histology) at the first visit or with a history of high-grade lesion were excluded from the study. Socio-demographic characteristics, virological, pathological and clinical data were retrieved from medical records.

All samples were stored into a biobank for which a declaration of preparation and storage of human samples for research use has been sent to the Ministère de l’Enseignement Supérieur et de la Recherche (no. DC-2014-2086).

### 2.2. Cytological and Histological Data

Cytological and histological samples were analyzed at the Pathology Department of the University Hospital of Besançon. The Bethesda System was used for reporting Pap smear results. The WHO 2014 classification was used to describe histological results.

### 2.3. High-Risk HPV DNA Testing

hrHPV DNA testing was performed from specimen transport medium (STM) using the clinically validated hc2 assay according to the manufacturer’s instructions. This assay detects the most common hrHPV types namely HPV 16, 18, 31, 33, 35, 39, 45, 51, 52, 56, 58, 59 and 68 but does not differentiate HPV types. Presence or absence of hrHPV DNA in the cervical specimens was defined according to the strength in relative light units (RLU) compared to the HPV positive control (PC) harboring 1 pg/mL of HPV16 DNA. Samples were considered positive when the RLU/PC ratio was 1.

All hrHPV positive samples were also tested for HPV16 and HPV18 VL. DNA was extracted from 100 µL of STM with the QIAamp DNA Mini Kit (Qiagen). Simplex real-time PCRs were used to quantify viral load with the LightCycler 480 (Roche Diagnostic, Meylan, France) using 2× Takyon qPCR kit (Kaneka Eurogentec, Seraing, Belgium). Primer and probe sequences are described in the Appendix A. Standard curves for viral DNA quantification were made with serial 1:10 dilutions over a range of 7 log concentrations of pBR322 HPV16 or pBR322 HPV18 plasmids. The thermal cycling conditions were a hot start for 3 min at 95 °C followed by 10 min at 95 °C and then 40 cycles of amplification (95 °C for 15 s, 60 °C for 1 min). Albumin gene quantification was used for normalization of VL considering that each cell harbors 2 albumin gene copies. Viral loads were expressed either as HPV16 and HPV18 genome equivalent (GE) per 10^3^ cells or as HPV16 and HPV18 GE per mL of cervical sample. For greater convenience and legibility, the VL values have been log_10_ transformed.

### 2.4. Data Analyses

Patients’ characteristics at baseline are shown for the study population. Mean with standard deviation (SD), median with interquartile range (IQR) and frequency with percentage were used to describe continuous and categorical variables, respectively, and compared with Wilcoxon test and Chi-Square test or Fisher test when necessary.

Median and 95% confidence interval (CI) for time-to-event analyses were estimated with the Kaplan–Meier method and compared with the log-rank test. Cox regression models were performed to estimate the hazard ratio (HR) with 95% CI. Association between major characteristics at baseline and time to high-grade lesion progression was assessed in Cox univariate analysis. Variables with *p* value < 0.1 in univariate analysis were introduced in the multivariate model and a stepwise selection was performed. Hazard proportionality was checked by plotting log-minus-log survival curves and colinearity between variables was assessed. Restricted cubic spline methodology was used to estimate the link between continuous variables and the outcome in order to identify an optimal cut-off of VL for categorization.

To identify populations with different risk of high-grade lesion occurrence, a prognostic score was proposed derived from the multivariable Cox model; for each variable, a weight of 0 was given to the reference modality and the HR rounded to the nearest integer for the other modalities, the sum of the weights giving the final score. Three groups of patients were thus identified namely at low risk (score = 0–1), intermediate risk (score = 2–3) and high risk (score 4–6).

All tests were two-sided and *p*-values were considered as statistically significant when <0.05. All analyses were performed using SAS version 9.3 (SAS Institute, Cary, NC, USA) and R software version 3.4.3 (R Development Core Team, Vienna, Austria; http://www.r-project.org, accessed on 14 June 2021).

## 3. Results

### 3.1. Population

The flow chart of the study is presented in Appendix A. Among the 10,601 women attending the Gynecology Clinic for the first time and co-tested with conventional Pap cytology and hrHPV test, 2102 (19.8%) had a positive hrHPV test with HPV 16/18 VL available. Four hundred eighty-three women with a cervical high-grade lesion at the first visit, 47 women having had a previous history of high-grade lesion and 687 women with no follow-up visit (FU) at the Gynecology Clinic of Besançon’s Hospital were excluded from the analysis. Finally, 885 patients (56.0%) having had ≤LSIL cytology at the first visit and at least one FU visit were referred as to the “study population”. Among them, 135 (15%) developed a high-grade lesion of the cervix during the FU.

The socio-demographic characteristics of the study population are described in Table 1. The mean age was 37 years. The majority of women (60%) had 1–3 children and oral contraception was used by 47% of the population. Thirty two percent of women were current smokers.

### 3.2. Pathological and Virological Characteristics at First, Visit

Equivocal and mildly abnormal cytology were detected in approximately 22% of samples (Table 2). The majority of cytological abnormalities were LSIL (16.5%), while ASC-US represented 5.7% of abnormal smears. Histological results were available for 11% of women and CIN1 was observed in 5.7% of cases.

The median value of RLU/PC given by the hc2 test was 53 (8–319) RLU/PC. Thirty percent of samples presented low (1–10 pg/mL) or middle (11–100 pg/mL) values of RLU/PC and 40% presented high RLU/PC values (>100 pg/mL). The overall HPV16 and HPV18 prevalence was 25.9% and 8.4%, respectively. HPV16 and HPV18 median VL were 3.1 and 3.5 log_10_GE/10^3^ cells, respectively.

At the end of the follow-up, a prevalence of HPV16 of 46.6% and 21.3% was observed in the subgroup of women with high-grade lesion and in the subgroup of women without high-grade lesion, respectively (*p* < 0.0001). The prevalence of HPV18 was 8.1% and 8.0% in the subgroup of women with high-grade lesion and in the subgroup of women without high-grade lesion, respectively (*p* = 0.992).

### 3.3. Follow-Up and Time to Lesion

The 885 patients included in the analysis had a median of 2 additional visits (from 1 to 11) with cervical samples taken for cytology and HPV testing. The median delay between two visits was 12 months. A high-grade lesion was diagnosed in 135 patients (15%) during the FU and several baseline variables were analyzed using the Cox univariate model (Table 3). Parity (*p* = 0.44) and immunodepression (*p* = 0.75) were not associated with a risk to develop a high-grade lesion. A trend for tobacco consumption to be associated with a higher risk for high-grade lesion occurrence was identified (HR 1.4 95% CI 0.99–1.99 *p* = 0.0582). Using local contraception or being menopausal proved to be protective for the occurrence of high-grade lesion (HR 0.62 95% CI 0.41–0.93 *p* = 0.0223).

As for cytology results, ASC-US or LSIL smear was significantly associated with a higher risk of high-grade lesion development compared to a Negative for Intraepithelial Lesion and Malignancy (NILM) smear (HR = 2.16; 95% CI 1.54–3.05 *p* < 10^−4^).

Furthermore, there was an association between the development of a high-grade lesion and the semi-quantitative hrHPV load determined by the hc2 assay (*p* < 0.0001). A significant increased risk (HR = 1.98 95% CI 1.41–2.78 *p* < 10^−4^) of high-grade lesion outcome was associated with >100 RLU/PC load compared to a 1–100 RLU/PC load. Since the presence of HPV16 DNA (Table 3) in baseline smears was associated with a 2.39 higher risk of high-grade lesion (*p* < 0.0001; 95% CI 1.70–3.37), the presence of HPV18 did not confer any increased risk.

Not surprisingly, Kaplan–Meier curves showed that the time to high-grade lesion occurrence was shorter in HPV16 positive women compared to those infected by hrHPVs other than HVPV16/18 (Figure 1). In contrast, the time to high-grade lesion occurrence was not different between women infected by HPV18 and women infected by hrHPV other than HPV16/18 (Figure 1).

Then, the prognostic value of the viral load assessed using hc2 assay was addressed in the limited group of HPV16+ cases. As shown in Appendix A, the time to CIN2+ occurrence was shorter in women with >100 RLU/PC load compared to those with a 1–100 RLU/PC load. Such a prognostic value was not observed in the subgroup of HPV18 positive women.

Restricted cubic spline analysis showed linear association for log_10_ of HPV16 VL in relation to high-grade lesion development. Thus, an HPV16 VL cut-off set at 3.2 log_10_GE/10^3^ cells permitted to identify a subgroup of HPV16 infected women at high risk of developing high-grade lesion with a HR of 3.09 (95% CI 2.14–4.48 *p* < 0.0001) (Table 3). Furthermore, the Kaplan–Meier estimate showed that HPV16 DNA load at first visit and lesion free probability were strongly correlated. Thus, women with a HPV16 VL below 3.2 log_10_GE/10^3^ cells (HPV16 low) had a significantly longer time to high-grade lesion probability compared to women presenting a HPV16 VL > 3.2 log_10_GE/10^3^ cells (HPV16 high) (Figure 2) (*p* = 0.0064). No specific HPV18 VL threshold could have been defined to predict the outcome of a CIN2+ (Table 3).

In multivariate analysis, HPV16 load (absence/log_10_GE/10^3^ cells <3.2 vs ≥3.2), RLU/PC (1–100 pg/mL vs >100 pg/mL) and cytology (normal vs abnormal) were independently associated with a significant increased risk of high-grade lesion development (Table 4) and were used to construct the prognostic score. In a Cox univariate model, scores 2–3 and 4–6 were significantly associated with a higher risk of high-grade lesion outcome with HRs of 2.39 (95% CI 1.61–3.57 *p* < 10^−4^) and 4.53 (95% CI 2.92–7.01 *p* < 10^−4^), respectively (Appendix A, Appendix A).

## 4. Discussion

In this study, we confirm on a large cohort of women infected by hrHPV and presenting without or with mild abnormalities that HPV16 DNA load may independently predict the development of CIN2+ [21,22,26,27,28,29,30,31]. On the contrary, being infected by HPV 18, did not confer an increased risk of high-grade cervical lesion development compared to other hrHPVs.

In this study, the high incidence of high-grade lesions is consistent with a previous study conducted in our hospital showing that 18.5% of hrHPV infected women with normal cytology or mild cytological abnormalities developed a high-grade lesion [4]. Now considering that hrHPV+ women infected by HPV16/18 with no of mild cervical abnormalities are at risk of cervical lesions, it cannot be excluded that some cases were already present at baseline since no colposcopy was done. As expected, Cox univariate analysis revealed that in this group of hrHPV infected women, an abnormal cytology was a prognostic factor associated with the evolution to high-grade lesions [4,32,33]. In the same line, RLU/PC values >100 increased significantly the risk of developing CIN2+ confirming previous observations and the potential usefulness of hrHPV load as a marker towards pre-cancers [4,34,35,36].

The proportion of HPV16 and HPV18 positive samples among hrHPV positive samples was similar in this cohort to that reported in French or international studies [7,8,37,38]. As expected, at the end of the follow-up the prevalence of HPV16 was higher (almost 2-fold higher) in women presenting a high-grade lesion than in women with no high-grade lesion. This observation confirms that HPV16 infections are at high risk of high-grade lesion development. Even if HPV16 and HPV18 are the most frequently detected genotypes in cervical cancers [6,7], HPV16 but not HPV18 infected women presented an increased risk of CIN2+ development. Numerous studies revealed that HPV18 was not consistently ranked at the second place after HPV16 among the most at-risk genotypes. In this series, the prevalence of HPV18 did not increase notably in CIN2+ cases compared to normal cytology. For instance, in a large series of hrHPV positive women with normal cytology the three years cumulative risk of CIN2+ was the highest for those infected by HPV16 (16.7%), followed by those infected by HPV52 and HPV31 (10.2%) and then by those infected by HPV18 [17]. Numerous other papers, reviewed by Cuzick and Wheeler [39], revealed that HPV31 and HPV33 were associated with a higher risk of CIN2+ lesion than HPV18. They also pointed out that HPV18 was more specifically linked to endocervical adenocarcinoma in situ and adenocarcinoma. Because complete genotyping is not available as a routine test in our hospital, individual genotype risk could not be assessed in the current study. Interestingly, high viral loads (>100 RLU/PC) assessed using the hc2 test add prognostic value in the subgroup of hrHPV positive patients infected by HPV16.

Next, the specific HPV16 VL was investigated as a risk factor enabling prediction to high-grade lesion. Kaplan–Meier curves for CIN2+ lesion-free probability clearly showed that a HPV16 load with a cut-off set at 3.2 log_10_GE/10^3^ cells was significantly associated with the development of histologically proven high-grade squamous intraepithelial lesion. Very scarce studies addressed the question of a HPV16 load cut-off as a prognostic factor for incident high-grade lesion. Using a normalized HPV16 load, Monnier et al. in our laboratory proposed a cut-off value of 200 GE/10^3^ cells for discriminating women with the highest risk for developing CIN2+ [21]. It is noteworthy that this cut-off is of the same order of magnitude (less than 1 log_10_ difference), even if the present study design was different. The multivariate analysis confirmed that HPV16 VL was an independent prognosis factor for high-grade lesion outcome as were RLU/PC values and cytology.

To go further, composite scores grouping these three variables were built. As expected, the HRs increased with the scores and reached a maximum for the highest scores. This is in line with recent data showing that a combination of virological parameters (hrHPV load assessed by hc2 and HPV16/18 genotyping), proved to be efficient to screen for prevalent CIN2+ [36].

The present survey conducted on a large cohort of women followed-up for 8 years provides strong evidence that viral load may be a relevant signature to efficiently triage women infected by HPV16 and presenting without or with equivocal or mildly abnormal cytology (ASC-US; LSIL) for the occurrence of high-grade lesion. From a clinical point of view, the present data obtained in the context of switching primary cervical cancer screening from cytology to population-based screening programs with hrHPV testing [13] are of most importance. Indeed, since hrHPV testing is a very sensitive strategy for CIN2+ screening, biomarkers are still needed to better triage women with the highest risk of cervical (pre-) cancer. In this context, measuring HPV16 load should be seriously considered. This may be particularly relevant in hrHPV-positive women with normal cytology that could be referred directly to colposcopy if a high HPV16 load (>3.2 log_10_GE/10^3^ cells) is measured.

Validation guidelines for specific HPV DNA load quantification are lacking and addressing the relevance of VL for other hrHPVs, notably HPV31, 33 or 52, remains an open question. If home-made HPV load quantification tests are adapted for research use, they may not be sufficiently robust to be used in a routine laboratory. Now, numerous commercial hrHPV screening tests are available that permit partial HPV16/18 identification [40]. Most of these in vitro diagnosis devices, dedicated to qualitative detection of HPV nucleic acids, are based on real-time PCR. It would be interesting to exploit their quantitative potential to assess HPV load directly from the primary sample. Furthermore, investigations are needed to prospectively evaluate the benefit of HPV DNA load in cervical cancer screening program. It is anticipated that such large studies would also help defining the best cut-offs for optimal management of hrHPV positive women with no or mild abnormalities at the cervix level.

The present study is based on virological and pathological data obtained prospectively in a routine medical care setting from patients attending the Gynecology Clinic of the Besançon’s University Hospital. This pragmatic design based on standard operating procedures for cervical cell sampling, sample transportation and laboratory analyses allow us to get close to real-life conditions. It also demonstrates the feasibility of implementing new biological parameters to improve cervical cancer screening. Validated real-time PCR were used for HPV16 and HPV18 DNA quantification [25], a technology that makes it possible to accurately and objectively evaluate VL. It also permits to set-up a clinically relevant cut-off for HPV16 load at 3.2 log_10_GE/10^3^ cells for patient’s risk stratification. Now, the hospital-based population of women included in this study is probably not representative of the general population. Thus, one should be cautious regarding the generalization of these results and it would be important to reproduce these results in an independent cohort. Furthermore, the impact of multiple infection on VL performance has not been addressed since HPV genotyping is not performed routinely.

## 5. Conclusions

In the case of hrHPV positivity, triage tests are needed to optimally manage women. In the present study, we confirm the clinical usefulness of HPV16 genotyping and HPV16 VL load to identify women most at risk of cervical high-grade lesion.

## Figures and Tables

**Figure 1 cancers-13-04149-f001:**
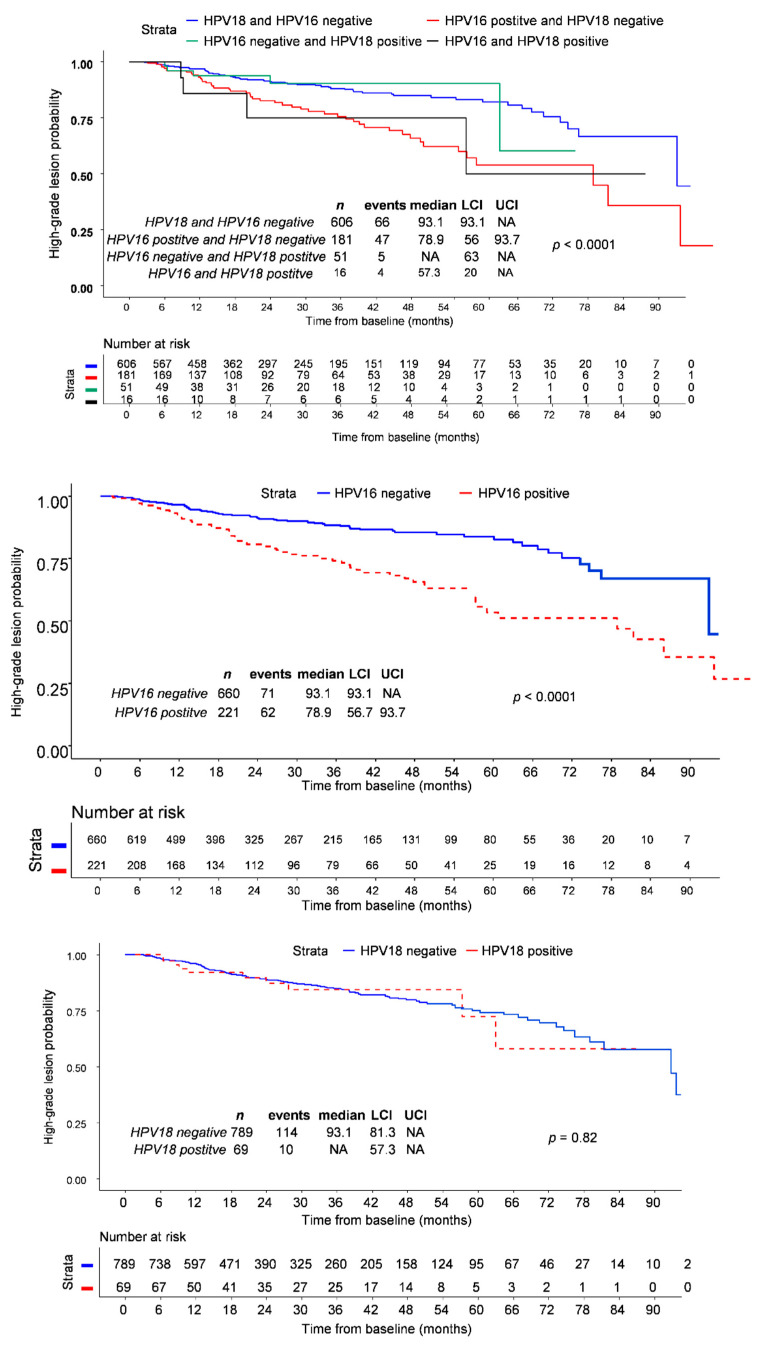
Kaplan-Meier estimates for time from baseline to the development of CIN2+ according to HPV16/HPV18 status (up), HPV16 (middle) and HPV18 (down) status vs other hrHPV. LCI: 95% Lower Confidence Interval; 95% UCI: Upper Confidence Interval. NA: Not available; HPV: Human Papillomavirus.

**Figure 2 cancers-13-04149-f002:**
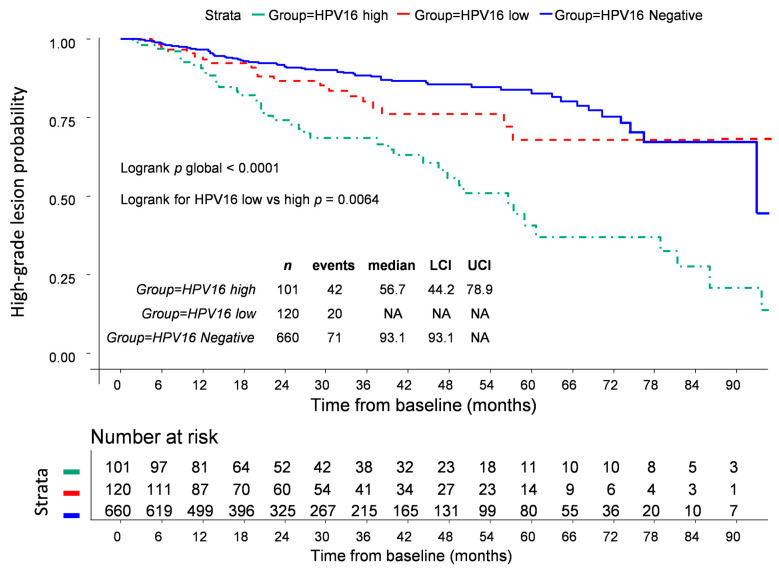
Kaplan-Meier estimates for time from baseline to the development of CIN2+ according to HPV16 viral load (log_10_GE/10^3^ cells). LCI: 95% Lower Confidence Interval; UCI: 95% Upper Confidence Interval. NA: Not available; HPV: Human Papillomavirus.

**Table 1 cancers-13-04149-t001:** Socio-demographic characteristics of the study population.

Characteristics	Study Population (*n* = 885) *n* (%)
**Age (year)**	
Mean ± SD	37.4 ± 11.7
Median (q1–q3)	34.5 (28.5–44.5)
**Parity**	
0	260 (35.0)
1	191 (25.7)
2–3	260 (35.0)
>3	33 (4.4)
NA	141
**Contraception**	
Oral contraception	381 (47.7)
Local contraception	138 (17.3)
Intra-uterine device	155 (19.4)
Menopause	124 (15.5)
NA	87
**Immunocompromised**	
Transplantation	21 (15.8)
HIV infection	34 (25.6)
Haemopathy and cancer	33 (24.8)
Auto-immune disease	44 (33.1)
Chronic renal failure	1 (0.7)
NA	752
**Tobacco smoking history**	
Current smoker	284 (32.1)
Never / Ex-smoker	601 (67.9)
**Sexually Transmitted Infection**	
HIV *	34 (47.2)
*Chlamydia*	24 (33.33)
HSV2	16 (22.2)
Syphilis	2 (2.8)
HBV	2 (2.8)
NA	813

SD: Standard deviation; NA: Not available; HIV: Human immunodeficiency virus; HSV2: Herpes simplex virus 2; HBV: Hepatitis B virus; * including co-infection cases as HIV+Chlamydia or HIV+Syphilis.

**Table 2 cancers-13-04149-t002:** Cytology, Histology, hrHPV, HPV16 and HPV18 DNA loads.

Variables	Study Population (*n* = 885) *n* (%)
**Cytology**	
Unsatisfactory	66 (7.5)
NILM	621 (70.3)
ASC-US	51 (5.7)
LSIL	146 (16.5)
NA	1
**Histology**	
Normal	45 (5.9)
CIN1	50 (5.7)
NA	790 (89.3)
**hc2**	
Mean RLU/PC (pg/mL) ± SD	326 ± 597
Median RLU/PC (pg/mL) (q1–q3)	53 (8–319)
**hc2 RLU/PC distribution**	
1–10 pg/mL	262 (29.6)
11–100 pg/mL	267 (30.2)
>100 pg/mL	356 (40.2)
**HPV16**	
Negative	660 (74.9)
Positive*	221 (25.9)
Mean (Log_10_GE/10^3^ cells) ± SD	3.1 ± 1.3
Median (q1–q3) (Log_10_GE/10^3^ cells)	3.2 (2.1–4)
Mean (Log_10_GE/mL) ± SD	7.1 ± 7.9
Median (q1–q3) (Log_10_GE/mL)	5.8 (5–6.6)
NA	
**HPV18**	
Negative	789 (92)
Positive *	69 (8.4)
Mean (Log_10_ GE/10^3^ cells) ± SD	3.5 ± 1.5
Median (q1–q3) (Log_10_GE/ 10^3^ cells)	3.6 (2.7–4.6)
Mean (Log_10_GE/ mL) ± SD	6.2 ± 1.4
Median (q1–q3) (Log_10_GE/mL)	6.2 (4.3–7.1)
NA	27

* including women for whom HPV16 or HPV18 viral loads were below the limit of quantification; ASC-US: Atypical Squamous Cells of Undetermined Significance; CIN: Cervical Intraepithelial Neoplasia; GE: Genome equivalent; hc2: Hybrid Capture 2; LSIL: Low-grade Squamous Intraepithelial Lesion; NA: Not Available; NILM: Negative for Intraepithelial Lesion or malignancy; RLU/PC: Relative Light Unit/Positive Control; SD: Standard Deviation

**Table 3 cancers-13-04149-t003:** Univariate Cox regression for time from baseline to the development of CIN2+ in the studied population with follow-up.

Variables	*n* (Events) ^a^	HR	95% CI	*p-*Value
**Parity**				
0	260 (44)	1	-	0.44
1	191 (32)	0.87	0.55–1.37	-
2–3	258 (45)	0.86	0.63–1.45	-
>3	33 (2)	0.32	0.08–1.33	-
**Contraception**				
Yes ^b^	536 (95)	1	-	0.0223
No ^c^	262 (30)	0.62	0.41–0.93	-
**Immunodepression**				
No	737 (112)	1	-	0.75
Yes	145 (23)	0.93	0.59–1.46	-
**Tobacco smoking**				
No	598 (85)	1	-	0.0582
Yes	284 (50)	1.4	0.99–1.99	-
**Cytology**				
Normal	621 (76)	1	-	<0.0001
Abnormal	263 (58)	2.16	1.54–3.05	-
**Hc2**				
1–100 pg/mL	529 (61)	1		<0.0001
>100 pg/mL	356 (74)	1.98	1.41–2.78	-
**HPV16**				
Absence	660 (71)	1	-	<0.0001
Presence	221 (62)	2.39	1.70–3.37	-
**HPV16 load**				
Absence and Log_10_ HPV16 GE/10^3^ cells < 3.2	780 (91)	1		<0.0001
Log_10_ HPV16 GE/10^3^ cells ≥ 3.2	101 (42)	3.09	2.14–4.48	-
**HPV18**				
Absence	789 (114)	1	-	0.83
Presence	69 (10)	1.7	0.56–2.05	-
**Log_10_ HPV18 GE/10^3^ cells**				
<5.2	59 (7)	1	-	0.0968
≥5.2	10 (3)	3.19	0.81–12.51	-

^a^ number of subject (number of CIN2+); ^b^ Oral contraception and intra-uterine device; ^c^ Local contraception and menopause; HR: hazard ratio; CI: confidence interval.

**Table 4 cancers-13-04149-t004:** Multivariate Cox regression for time from baseline to the development of high-grade lesion in the studied population with follow-up.

Variables	*n* (Events)	HR	95% CI	*p-*Value	Weight
-	880 (132)	-	-	-	-
**Hc2**					
1–100 pg/mL	527 (60)	1	-	0.05	0
>100 pg/mL	353 (72)	1.44	1.00–2.08	-	1
**Cytology**					
Normal	618 (75)	1	-	<0.0001	0
Abnormal	262 (57)	2.02	1.43–2.87	-	2
**HPV16**					
Absence and Log_10_ HPV16 GE/10^3^ cells < 3.2	780 (91)	1	-	<0.0001	0
Log_10_ HPV16 GE/10^3^ cells ≥ 3.2	100 (41)	2.67	1.80–3.97	-	3

HR: hazard ratio; CI: confidence interval.

## Data Availability

Data are available upon request to the corresponding author.

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
