# Peer review of "HPV16 Load Is a Potential Biomarker to Predict Risk of High-Grade Cervical Lesions in High-Risk HPV-Infected Women: A Large Longitudinal French Hospital-Based Cohort Study"

_cancers, 2021, doi:10.3390/cancers13164149_

Round 1

Reviewer 1 Report

The paper by Baumann et al. entitled HPV16 and HPV18 Viral Loads as Potential  Biomarkers to Predict Risk of High-Grade Cervical Lesions: a Large Longitudinal French Hospital-Based Cohort Study reports the results of a study designed to assess the value of HPV16 and HPV18 viral loads as predictors of the development of cervical high-grade lesion. In a case panel of 885 cytological specimens with no or mild cytological abnormalities and found positive using the Hybrid Capture 2 (hc2) assay, HPV16 and HPV18 viral loads were further assessed using a PCR assay. The prevalence of HPV16 and HPV18 was 25.9% and 32 8.4% respectively. During the follow-up, 15% of those women developed a high-grade lesion. Data analysis found that the presence of cytological abnormalities (RR 2.16), HPV16 positivity (RR 2.39) and high HPV16 load (RR 3.09) were associated with increased risk of high grade lesion. No significant increased risk was observed for HPV18 positive cases.

The study is based on a large population of patients, the methodology is sound and the authors have a large experience in the field. The Figure 2 provides unequivocal data on the impact of HPV16 viral load on the development of high grade lesion. The paper is well written and deserves publication.

Remarks

  • Since no significant link was observed between the presence of HPV18 or a high HPV18 load and the development of high grade lesion, it is questionable to mention HPV18 load as a potential biomarker of the disease outcome in the title. This lack of association may be related to the small number (69) of HPV18 cases as compared with the 221 HPV16 positive cases. It would be interesting to know what the figure 1 (down) would be if the comparison had been performed between HPV18 and non HPV16/18 cases. A significant difference, or at least a trend of association would justify keeping HPV18 in the title.
  • In practice, it would be poorly realistic to assess viral load via a supplementary biological test. However, the report indicates that, using HC2 assay, a significant increased risk of high-grade lesion outcome (HR=1.98) was associated with >100 RLU/PC load compared to a 1-100 RLU/PC load. Since in the present work the HPV16 cases have been determined by a supplementary PCR test, it would be interesting to show what the prognostic value of the viral load assessment using HC2 assay would be in the limited group of HPV16+ cases. The authors mention in the discussion that screening tests are available that permit HPV16/18 identification. In the perspective to determine both viral genotype and viral load using the screening test, the present study would be reinforced by providing preliminary results suggesting that viral load assessment using theHC2 test is able to add prognostic value in patients with HPV16 positive test.
  • In clinical practice, patients with abnormal cytology and positive viral test benefit from a direct colposcopy, whereas patients with normal cytology and positive viral test are referred for another viral test after one year of follow-up. This later group (false negative of cytology or viral infection without cytopathogenic effect) would benefit mostly from a test indicating the risk of development of a high grade lesion. Therefore, in the present study, it would be interesting to assess this risk in the group of patients with normal cytology/HPV16/high viral load and to provide the mean time of progression to high grade SIL. This would allow assessing the potential benefit of performing a colposcopy at once VS after a delay of one year in patients with “biological high risk”.
  • The beginning of the Discussion paragraph is redundant with the Results It could be shortened to leave space to add comments on the additive value of the present work in comparison with other similar reports and to discuss the potential impact of viral load assessment in clinical practice (cf precedent comment).
  • Please check the figure of 56.0% for the study population (paragraph 3.1 and supplementary data). Could it be 42.1% (885/2012) ? or justify.

Author Response

The paper by Baumann et al. entitled HPV16 and HPV18 Viral Loads as Potential Biomarkers to Predict Risk of High-Grade Cervical Lesions: a Large Longitudinal French Hospital-Based Cohort Study reports the results of a study designed to assess the value of HPV16 and HPV18 viral loads as predictors of the development of cervical high-grade lesion. In a case panel of 885 cytological specimens with no or mild cytological abnormalities and found positive using the Hybrid Capture 2 (hc2) assay, HPV16 and HPV18 viral loads were further assessed using a PCR assay. The prevalence of HPV16 and HPV18 was 25.9% and 32 8.4% respectively. During the follow-up, 15% of those women developed a high-grade lesion. Data analysis found that the presence of cytological abnormalities (RR 2.16), HPV16 positivity (RR 2.39) and high HPV16 load (RR 3.09) were associated with increased risk of high grade lesion. No significant increased risk was observed for HPV18 positive cases.

The study is based on a large population of patients, the methodology is sound and the authors have a large experience in the field. The Figure 2 provides unequivocal data on the impact of HPV16 viral load on the development of high grade lesion. The paper is well written and deserves publication.

The authors would like to thank the reviewer for this positive assessment of our work.

Remarks

  • Since no significant link was observed between the presence of HPV18 or a high HPV18 load and the development of high grade lesion, it is questionable to mention HPV18 load as a potential biomarker of the disease outcome in the title. This lack of association may be related to the small number (69) of HPV18 cases as compared with the 221 HPV16 positive cases. It would be interesting to know what the figure 1 (down) would be if the comparison had been performed between HPV18 and non HPV16/18 cases. A significant difference, or at least a trend of association would justify keeping HPV18 in the title.

We agree with the reviewer that the lack of significant link between the presence of HPV18 and the development of high grade lesion may be due to the small number of HPV18 cases. Comparison between HPV18 and non HPV16/18 cases is now presented in figure 1a. As shown in the Kaplan-Meier estimates for the development of CIN2+, HPV18 cases do not behave differently compared to HPV16 negative/HPV18 negative cases. Therefore, HPV18 is probably not a potential biomarkers in this series. Consequently, the title of the manuscript has been changed.

  • In practice, it would be poorly realistic to assess viral load via a supplementary biological test. However, the report indicates that, using HC2 assay, a significant increased risk of high-grade lesion outcome (HR=1.98) was associated with >100 RLU/PC load compared to a 1-100 RLU/PC load. Since in the present work the HPV16 cases have been determined by a supplementary PCR test, it would be interesting to show what the prognostic value of the viral load assessment using HC2 assay would be in the limited group of HPV16+ cases. The authors mention in the discussion that screening tests are available that permit HPV16/18 identification. In the perspective to determine both viral genotype and viral load using the screening test, the present study would be reinforced by providing preliminary results suggesting that viral load assessment using theHC2 test is able to add prognostic value in patients with HPV16 positive test.

As suggested by the reviewer, the prognostic value of the hc2 load was assessed in the limited HPV16+ cases. As shown in the result section, it was confirmed that the viral load determined using the hc2 test add prognostic value in HPV16 positive patients. This is now reported in the result section and discussed.

  • In clinical practice, patients with abnormal cytology and positive viral test benefit from a direct colposcopy, whereas patients with normal cytology and positive viral test are referred for another viral test after one year of follow-up. This later group (false negative of cytology or viral infection without cytopathogenic effect) would benefit mostly from a test indicating the risk of development of a high grade lesion. Therefore, in the present study, it would be interesting to assess this risk in the group of patients with normal cytology/HPV16/high viral load and to provide the mean time of progression to high grade SIL. This would allow assessing the potential benefit of performing a colposcopy at once VS after a delay of one year in patients with “biological high risk”.

We would like to thank the reviewer for this interesting remark. The risk of development of a high grade lesion was assessed in the subgroup of women with normal cytology and high viral load.  As shown in the result section, women with HPV16 load >3.2 log10eg/103 cells were at higher risk of CIN2+ development compared to women with lower HPV16 VL. As suggested by the reviewer, these women should probably benefit of performing a colposcopy at once. This has been added in the discussion.

  • The beginning of the Discussion paragraph is redundant with the Results It could be shortened to leave space to add comments on the additive value of the present work in comparison with other similar reports and to discuss the potential impact of viral load assessment in clinical practice (cf precedent comment).

The beginning of the discussion paragraph has been modified accordingly.

  • Please check the figure of 56.0% for the study population (paragraph 3.1 and supplementary data). Could it be 42.1% (885/2012) ? or justify.

We thank the reviewer for highlighting this mistake. This has been corrected.

Reviewer 2 Report

Overall, the idea and the message are very valuable, but the results require a bioinformatics analysis, including the use of appropriate tests and models. The discussion should be edited and adapted to the results obtained.

Selected comments:

Lines 8-16 although French is a very beautiful language, proper names are in English

Lines 109  - a brief description and explanation of the reason for designing HPV 16/18 tests

Figures 1 and 2: add descriptions of the abbreviations LCI and UCI.

Figures 1 and 2: on the Y axis is the probability of high-grade lesion, not the time.

What hypothesis is tested in multiple regression? Please mention which variables were tested for that hypothesis. It is a mistake to use 2 very similar tests (HPV16 and hc2) in The Cox proportional-hazards model, since HPV16 is a subgroup of the hc2 test. In my opinion, only 1 test should be selected in the analysis. Multiple analysis must be performed with an appropriate selection of variations and a properly performed interpretation.

Table 3 : what is "n" and what is "events"? Table 3 should be improved, the column “n(events)” is not clear.

Line 321-324: There is no basis for the following conclusions: “Thanks to very good clinical performances (high clinical sensitivity, high negative predictive value), it allows to safely extend screening intervals when the test is negative”. Sensitivity, specificity, negative predictive value and positive predictive value calculations were not presented in the results.

Line 235 -236 “On the contrary, being infected by HPV 18, did not confer an increased risk of high-grade cervical lesion development compared to other hrHPVs. “–Please show this result in the "results" section, in my opinion there is no such comparison.

Lines 260. The percentage of HPV16 in the CIN2 + subgroup and the percentage of HPV18 in the CIN2 + subgroup should be reported and discussed. You should also specify the percentage of HPV16 in the "normal" subgroup and the percentage of HPV18 in the normal subgroup, analyze with the chi2 test, give the OD (odd ratio) and discuss it.

Lines 279 - 280 “The multivariate analysis confirmed that HPV16 VL was an independent prognosis factor for high-grade lesion outcome as were RLU/PC values and cytology.” -The multiple analysis must be repeated, in addition, if in such an analysis three features are revealed, it does not speak of an independent factor, but a subgroup of cases who have these three features simultaneously.

Author Response

Overall, the idea and the message are very valuable, but the results require a bioinformatics analysis, including the use of appropriate tests and models. The discussion should be edited and adapted to the results obtained.

The authors would like to thanks the reviewer for his comments that helped us to improve the quality of our manuscript.

Selected comments:

Lines 8-16 although French is a very beautiful language, proper names are in English

This has been corrected.

Lines 109 - a brief description and explanation of the reason for designing HPV 16/18 tests

This is now justified. HPV16 and HPV18 are the most prevalent HPV in cervical cancers.

Figures 1 and 2: add descriptions of the abbreviations LCI and UCI.

Abbreviations are now defined in the figures.

Figures 1 and 2: on the Y axis is the probability of high-grade lesion, not the time.

This has been corrected.

What hypothesis is tested in multiple regression? Please mention which variables were tested for that hypothesis. It is a mistake to use 2 very similar tests (HPV16 and hc2) in The Cox proportional-hazards model, since HPV16 is a subgroup of the hc2 test. In my opinion, only 1 test should be selected in the analysis. Multiple analysis must be performed with an appropriate selection of variations and a properly performed interpretation.

While the hc2 test detects the presence of 13 hrHPV, the HPV16 test detect and quantify only one genotype. Now, it is true that the 2 tests are not independent. Indeed, the RLU/PC measured by the hc2 test in the subgroup of HPV16 positive women are different from those measured in the subgroup of HPV16 negative women (Wilcoxon test, p=0.0011). Nevertheless, the correlation between the viral load measured by the hc2 test and the HPV16 load is poor (Pearson correlation: 0.3). Moreover, as shown in figure S3, high RLU/PC values (>100 RLU/PC) clearly add prognostic value in HPV16 positive women. This is also the case for HPV16 negative women (please see attached document). Thus we feel that these two parameters bring complementary information to better identify patients with a high risk of CIN2+. This is why these two parameters were included in the multivariate analysis.

As for the hypothesis tested in multiple regression, the Cox regression models the association between different covariates and the time to high grade lesion. For each variable, the null hypothesis of the parameters was tested. The conditions of application of the model and in particular the proportional risk assumption of the model were tested as described in the method section. For the selection of variables, several choices can be made: stepwise, lasso, full model, etc... For this study, we chose a standard, parsimonious method, and only the variables statistically significant in univariate analysis were introduced in multivariate analysis.

Table 3 : what is "n" and what is "events"? Table 3 should be improved, the column “n(events)” is not clear.

“n” corresponds to the number of subjects; “events” refers to the number of CIN2+. This is now indicated in the table.

Line 321-324: There is no basis for the following conclusions: “Thanks to very good clinical performances (high clinical sensitivity, high negative predictive value), it allows to safely extend screening intervals when the test is negative”. Sensitivity, specificity, negative predictive value and positive predictive value calculations were not presented in the results.

This sentence is not the conclusion of the study but refers to a general assertion regarding hrHPV testing. The sentence has been modified to avoid any confusion.

Line 235 -236 “On the contrary, being infected by HPV 18, did not confer an increased risk of high-grade cervical lesion development compared to other hrHPVs. “–Please show this result in the "results" section, in my opinion there is no such comparison.

We agree with the reviewer that the comparison between HPV18 + women and those infected by other hrHPV had not been done. As shown in the new figure 1, the impact of HPV18 in hrHPV positive women has been assessed. No significant difference was observed when HPV18 positive women are compared with non-HPV18/HPV16 hrHPV-positive women. We therefore conclude that the major risk for high grade lesion development is due to HPV16 in this series.

Lines 260. The percentage of HPV16 in the CIN2 + subgroup and the percentage of HPV18 in the CIN2 + subgroup should be reported and discussed. You should also specify the percentage of HPV16 in the "normal" subgroup and the percentage of HPV18 in the normal subgroup, analyze with the chi2 test, give the OD (odd ratio) and discuss it.

As suggested by the reviewer, the percentage of HPV16 and HPV18 are now presented in CIN2+ and in the “normal” subgroup. These results have been briefly discussed.

Lines 279 - 280 “The multivariate analysis confirmed that HPV16 VL was an independent prognosis factor for high-grade lesion outcome as were RLU/PC values and cytology.” -The multiple analysis must be repeated, in addition, if in such an analysis three features are revealed, it does not speak of an independent factor, but a subgroup of cases who have these three features simultaneously.

We agree with the reviewer that such analysis should be repeated in an independent cohort. This has been discussed as a potential limitation of the study.

Round 2

Reviewer 2 Report

Again: Figure 2: on the Y axis is the probability of high-grade lesion, not the time.

English grammar to improve and / or explain what the authors meant in the following sentenses: “Not surprisingly, Kaplan Meier curves showed that the time to high-grade lesion occurrence was shorter in HPV16 positive women (either or not co-infected by HPV18) compared to those infected by other hrHPVs (Figure 1). This was not the case in hrHPV positive women infected by HPV18. but not by HPV16 who behave similarly to those infected by other hrHPV (green curve up panel)”.

Please enter the value of p for these calculations: “HPV16 and HPV18 prevalence were 21% and 7.8% in normal smears taken at inclusion. In CIN2+ cases HPV16 and HPV18 prevalence were 52.3% and 10.3% respectively”.

Conclusions: Again, there is no basis for the following conclusions: “hrHPV testing with validated tests has been introduced as the primary screening tool for cervical cancer in several countries and thanks to very good clinical performances (high clinical sensitivity, high negative predictive value), it allows to safely extend screening intervals when the test is negative”. Specificity, negative predictive value and positive predictive value calculations were not presented in the results (https://en.wikipedia.org/wiki/Sensitivity_and_specificity).

Please change sentence in abstract …. “Furthermore a composite score grouping HPV16 load, cytology and hrHPV detection permitted to stratify risk of CIN2+ development” …. according to the obtained results: “The results show the following: “In multivariate analysis, HPV16 load (absence/ log10ge/103 cells <3.2 vs ³3.2), RLU/PC 239 (1-100pg/mL vs >100pg/mL), and cytology (normal vs abnormal) were independently associated with a significant increased risk of high-grade lesion development (Table 4) and were used to construct the prognostic score”.

Author Response

Again: Figure 2: on the Y axis is the probability of high-grade lesion, not the time.

We apologize for this omission. This has been corrected.

English grammar to improve and / or explain what the authors meant in the following sentenses: “Not surprisingly, Kaplan Meier curves showed that the time to high-grade lesion occurrence was shorter in HPV16 positive women (either or not co-infected by HPV18) compared to those infected by other hrHPVs (Figure 1). This was not the case in hrHPV positive women infected by HPV18. but not by HPV16 who behave similarly to those infected by other hrHPV (green curve up panel)”.

The sentence has been corrected as follows:

Not surprisingly, Kaplan Meier curves showed that the time to high-grade lesion oc-currence was shorter in HPV16 positive women compared to those infected by hrHPVs other than HVPV16/18 (Figure 1). In contrast, the time to high-grade lesion occurrence was not different between women infected by HPV18 and women infected by hrHPV other than HPV16/18 (Figure 1).

Please enter the value of p for these calculations: “HPV16 and HPV18 prevalence were 21% and 7.8% in normal smears taken at inclusion. In CIN2+ cases HPV16 and HPV18 prevalence were 52.3% and 10.3% respectively”.

Probably we did not understand well the previous comment of the reviewer to calculate HPV16/18 prevalence. Actually at the end of the follow-up, a prevalence of HPV16 of 46.6% and 21.3% was observed in the subgroup of women with high-grade lesion and in the subgroup of women without high-grade lesion respectively (p<0.0001). The prevalence of HPV18 was 8.1% and 8.0% in the subgroup of women with high-grade lesion and in the subgroup of women without high-grade lesion respectively (p=0.992). As requested by the reviewer these results has been reported in the result section.

 However, we would like to point out that in this study, we made the choice to perform a survival analysis to model high-grade lesion occurrence. This approach allows to take into account all available information about high-grade lesions: whether it occurred or not and when. In our survival analysis, the data of patients without high-grade lesion during follow-up, even if they are lost of follow-up prematurely, is integrated in the estimation, what would not be the case with logistic regression at specific time point (1 or 2 years for example). Because of heterogeneity in follow-up between patients, we privileged hazard ratio (HR=2.39 (95%CI 1.70-3.37) for HPV16) instead of odd ratio estimated with a logistic regression at the end of follow-up or at a specific time point.

Conclusions: Again, there is no basis for the following conclusions: “hrHPV testing with validated tests has been introduced as the primary screening tool for cervical cancer in several countries and thanks to very good clinical performances (high clinical sensitivity, high negative predictive value), it allows to safely extend screening intervals when the test is negative”. Specificity, negative predictive value and positive predictive value calculations were not presented in the results (https://en.wikipedia.org/wiki/Sensitivity_and_specificity).

This sentence has been deleted.

Please change sentence in abstract …. “Furthermore a composite score grouping HPV16 load, cytology and hrHPV detection permitted to stratify risk of CIN2+ development” …. according to the obtained results: “The results show the following: “In multivariate analysis, HPV16 load (absence/ log10ge/103 cells <3.2 vs ³3.2), RLU/PC 239 (1-100pg/mL vs >100pg/mL), and cytology (normal vs abnormal) were independently associated with a significant increased risk of high-grade lesion development (Table 4) and were used to construct the prognostic score”.

The sentence has been changed as suggested by the reviewer.